# The Role of TRPM7 in Oncogenesis

**DOI:** 10.3390/ijms25020719

**Published:** 2024-01-05

**Authors:** László Köles, Polett Ribiczey, Andrea Szebeni, Kristóf Kádár, Tibor Zelles, Ákos Zsembery

**Affiliations:** 1Department of Oral Biology, Semmelweis University, H-1089 Budapest, Hungary; koles.laszlo@dent.semmelweis-univ.hu (L.K.); aszebeni@gmail.com (A.S.); kadar.kristof@dent.semmelweis-univ.hu (K.K.); zelles.tibor@dent.semmelweis-univ.hu (T.Z.); 2Department of Pharmacology and Pharmacotherapy, Semmelweis University, H-1089 Budapest, Hungary; 3Laboratory of Molecular Pharmacology, Institute of Experimental Medicine, H-1083, Budapest, Hungary

**Keywords:** TRPM7 channel, chanzyme, tumor hallmarks, cancer, malignancy, neoplasm

## Abstract

This review summarizes the current understanding of the role of transient receptor potential melastatin-subfamily member 7 (TRPM7) channels in the pathophysiology of neoplastic diseases. The TRPM family represents the largest and most diverse group in the TRP superfamily. Its subtypes are expressed in virtually all human organs playing a central role in (patho)physiological events. The TRPM7 protein (along with TRPM2 and TRPM6) is unique in that it has kinase activity in addition to the channel function. Numerous studies demonstrate the role of TRPM7 chanzyme in tumorigenesis and in other tumor hallmarks such as proliferation, migration, invasion and metastasis. Here we provide an up-to-date overview about the possible role of TRMP7 in a broad range of malignancies such as tumors of the nervous system, head and neck cancers, malignant neoplasms of the upper gastrointestinal tract, colorectal carcinoma, lung cancer, neoplasms of the urinary system, breast cancer, malignant tumors of the female reproductive organs, prostate cancer and other neoplastic pathologies. Experimental data show that the increased expression and/or function of TRPM7 are observed in most malignant tumor types. Thus, TRPM7 chanzyme may be a promising target in tumor therapy.

## 1. TRP Channels

The transient receptor potential (TRP) superfamily consists of non-selective cation-permeable channels, of which 28 polymodal ion channels have already been identified in mammals [1,2,3,4]. The TRP superfamily can be divided into seven subgroups based on sequence homology and membrane topology: the TRP-canonical (TRPC), TRP-vanilloid (TRPV), TRP-melastatin (TRPM), TRP-polycystin (TRPP) TRP-Mukolipin (TRPML), TRP-Ankyrin (TRPA) and TRP-NOMPC (TRPN; “N” stands for no mechanoreceptor potential C) ion channel subfamilies [4,5,6]. In general, TRP proteins function as ion channels in the plasma membrane, play a key role in cellular ion homeostasis, and often have a sensory function as well. TRP channels are expressed in all human organs and appear to play a central role in the pathophysiological processes of many diseases including carcinogenesis. An increasing amount of evidence indicates the connection between the various hallmarks of cancer and the altered expression/activity of multiple TRP channels [4,7,8,9,10,11,12,13,14,15,16]. TRP channels are composed of four subunits in a homomeric or hetero-oligomeric formation to create cation-selective channels [5,6,17]. Each TRP ion channel subunit has a transmembrane domain (TMD), consisting of six segments (S1–S6) with a pore-forming loop (P-loop) between S5 and S6 [4,18]. The intracellularly localized carboxyl and amino termini vary widely in amino acid sequence and length and contain domains and motifs that have a role in channel assembly, activation and functional regulation through protein–protein and protein–ligand interactions [18]. Members of the TRP superfamily are Na^+^- and K^+^-permeable, and some are also permeable to divalent cations, such as Ca^2+^ and Mg^2+^. TRP ion channels play a fundamental role in cellular sensation, which includes the detection of changes in the local extracellular environment, as well as the perception of external stimuli via various sensory modalities, such as vision, olfaction, hearing, taste, temperature and touch [4,5,19]. TRP channels also have a key role in ion homeostasis regulating bone remodeling, muscle contraction and vasomotor control [20]. The permeability for different monovalent and divalent cations varies among the TRP channels [3,21] and the activity of TRP channels can be influenced by various stimuli and binding partners [5]. The opening of TPR channels can depolarize the cell membrane due to the influx of cations, which in turn results in various cellular effects, e.g., neurotransmitter release, muscle contraction, gene transcription, cell proliferation and cell death [5,22]. The activation of Ca^2+^-permeable TRP channels leads to an increase in [Ca^2+^]_i_, orchestrating various physiological and pathological processes controlled by the intracellular Ca^2+^ signaling machinery. An increasing body of evidence links altered Ca^2+^ signaling and dysregulated TRP channel expression/activity to carcinogenesis and to various cancer hallmarks including tumor cell growth, survival, migration, invasion and metastasis or cell death [7,23,24].

The TRPM family represents the largest and most diverse group in the TPR superfamily [25,26]. This subfamily consists of eight ion channel proteins: TRPM1-8 with widespread expression in the human organs. Based on sequence homology, the TRPM subfamily is classified into four subgroups: TRPM1/3, TRPM2/8, TRPM4/5 and TRPM6/7 [25,27]. These subgroups are characterized by distinct ion permeability. For instance, while TRPM6/7 is highly permeable, TRPM4/5 is impermeable to Ca^2+^ and the latter proteins act as Ca^2+^-activated monovalent cation channels that functionally conduct Na^+^ ions [5,6,28,29].

The common structure of TRPM shares the overall architecture of TRP channels. It contains TMDs consisting of six helices (S1–S6) and the intracellularly localized N- and C-terminal sequences [4,18,25]. The TMD can be further divided into the ion-conducting pore domain, which consists of the transmembrane helixes S5 and S6 and the P-loop that is located between these two segments, and the S1–S4 domain, which surrounds the ion-conducting pore. A linker sequence between S4 and S5 connects the above-mentioned two domains, and it is thought to have fundamental role in the gating of the TRPM channels. The S1–S4 domain contains the binding sites of agonists (e.g., Ca^2+^) and other ligands, while in voltage-gated TRP ion channels this domain contributes to voltage sensing [25]. The C-terminal region includes the highly conserved TRP helix, a hallmark of TRP channels, the coiled-coil region and a C-terminal domain (CTD) that is highly variable among the several TRPM channels. Structurally the coiled-coil region can be divided into the coiled-coil “pole” and the helical “ribs” segments. The intracellular N-terminal sequence contains the TRPM major homology region (MHR) that forms two domains: the TRPM homology region 3/4 (MHR3/4) and 1/2 (MHR1/2) domains. The N-terminal MHR is thought to play a privileged role in channel assembly and trafficking [1,30]. In the channel architecture, the MHR domains in the TRPM channels form a large hollow that is penetrated by the C-terminal coiled-coil “pole” and “ribs” regions. The two C-terminal coiled-regions form a unique umbrella-like shape that contributes to the assembly of TRPM subunits into the tetrameric functional form [25]. Three channel proteins, TRPM2, 6 and 7, are particularly distinct among the TRPM family due to their uncommon enzymatic activities. On the C-terminal, TRPM2 contains an ADP-ribose pyrophosphatase domain (NUDT9 homology domain), whereas both TRPM6 and TRPM7 have serine-threonine protein kinase domains (α-kinase domains) [1,17]. TRPM ion channels are involved in various physiological processes, e.g., active intestinal absorption and renal reabsorption of Mg^2+^, the regulation of core body temperature, the modulation of insulin secretion and the regulation of cardiac conduction. TRPM channels further play a prominent role in various types of sensory functions, including cellular reduction–oxidation sensations, heat and taste sensations, as well as the perception of inflammatory pain [6,31].

## 2. The TRPM7 Chanzyme

TRPM7 (ChaK1, TRP-PLIK, and LTRPC7) protein is ubiquitously expressed in the human body. This protein is known to act as a chanzyme, since it has dual activity in its functional tertrameric composition. It is an ion channel allowing the flux of divalent cations (Ca^2+^, Mg^2+^, Zn^2+^, Mn^2+^, and Co^2+^); on the other hand, it acts as a serine-threonine protein kinase (α-kinase) [2,17,32,33]. Structurally, TRPM7 shares common features with other members of the TRPM subfamily: it has a TMD composed of six transmembrane helixes and the P-loop, an intracellularly localized N-terminal sequence that includes the TRPM homology region and a large cytosolic C-terminus that contains the TRP helix, the coiled-coil domain [25,33,34] (Figure 1). The presence of a C-terminal α-kinase domain in TRPM7 (similar to its closest homolog, TRPM6) endows the protein with kinase activity as well. Due to the dual function of the protein, it is also called the “TRPM7 chanzyme”. TRPM7 can phosphorylate serine/threonine residues of intracellular targets, including annexin 1 [35], myosin IIA, IIB, and IIC [36], histone H3 [37], RhoA [38] or Smad2 [39].

The role of TRPM7 has been repeatedly demonstrated in the regulation of Ca^2+^ and Mg^2+^ homeostasis, cell differentiation, cell viability, cell death and migration [33,40,41,42,43,44,45]. Besides the central role of TRPM7 in physiological functions, it is also linked to pathological processes such as oncogenesis, cancer progression, tumor cell survival and metastasis [6,27,33,46,47,48,49,50,51].

## 3. Role of TRPM Channels in Cancer

An increasing amount of evidence indicates the role of various TRPM proteins in pathophysiological processes related to carcinogenesis and cancer progression (Figure 2). Several studies demonstrate changes in mRNA and/or protein expression profiles for different TRPM channels in the various cancer types. Furthermore, data show that manipulations of TRPM channel expression and/or activity either via protein knockdown (silencing small interfering RNA, siRNA; short hairpin RNA, shRNA) [52,53] or using TRPM channel inhibitors [54,55] can significantly modify several hallmarks of cancer in the different tumor types [24,27,30,56,57,58,59,60]. For instance, in case of breast cancer, the expression of the TRPM2 channel is higher in both invasive and non-invasive tumors than in normal tissues. It has been also shown that TRPM2 has a protective role in breast cancer cells as it facilitates the integrity of genomic DNA by helping to minimize DNA damage. Nevertheless, TRPM2 promotes tumor growth in these cells [61,62,63]. The TRPM8 channel also influences breast cancer hallmarks. It is overexpressed in breast adenocarcinoma cells correlating particularly with estrogen receptor-positive tumors [52,64,65]. TRPM8 seems to be directly involved in invasiveness and metastasis by regulating epithelial–mesenchymal transition (EMT) [65,66]. Furthermore, increased TRPM7 expression appears to play a key role in various characteristics of breast cancer cells, e.g., increased proliferation, altered cell adhesion, migration and invasion potential [27,67,68].

## 4. Role of TRPM7 Chanzyme in the Pathophysiology of Various Tumors

Subsequent to the introduction of TRPM-related tumorigenesis, the following chapter focuses on TRPM7.

### 4.1. Tumors of the Nervous System

#### 4.1.1. Glioblastoma

Numerous studies demonstrated the role of TRPM channels, including TRPM7, in the tumorigenesis of glioblastomas. The data have been summarized in excellent recent reviews [50,83]. Functional expression of TRPM7 channels in human glioblastoma cells has been reported by several groups. The subsequently increased proliferation, migration and invasion of tumor cells were abolished via the knockdown or pharmacological inhibition of TRPM7 in various rodent and human glioblastoma cell lines. the inhibition of phosphatidylinositol 3-kinase (PI3K)/protein kinase B (Akt) and extracellular signal-regulated kinase kinase (MEK)/mitogen-activated protein (MAP) kinase signaling has been implicated in the observed effects [84,85,86,87,88,89,90,91,92]. Liu et al. have demonstrated that TRPM7 activates Janus kinase 2/signal transducer and the activator of transcription 3 (JAK2/STAT3) and/or Notch signaling pathways, resulting in the activation of aldehyde dehydrogenase 1 (ALDH1) promoters and increased proliferation, migration and invasion in a human glioblastoma cell line [93]. More recently, a detailed analysis has revealed that in TRPM7-mediated Notch1 signaling, the expression of CD133 and ALDH1 are crucial in glioma cell proliferation and glioma stem cell stemness. Unlocking G1/S arrests, the stimulation of cell entry into S and G2/M phases and inhibition of glioma cell apoptosis are also important factors in the sequence of events [94]. Wan et al. have proposed that the TRPM7 channel function was required for cell growth and proliferation while kinase activity was involved in increased migration and invasion. Furthermore, TRPM7 was shown to be a negative regulator of the tumor suppressor microRNA (miR)-28-5p [95]. A recent publication has revealed that the TRPM7/HOX transcript antisense intergenic RNA axis is overexpressed in glioma cells promoting cell proliferation and invasion. These oncogenic effects have been (at least in part) mediated by the downregulation of miR-301a-3p (an inhibitor of glioma cell growth and invasion) resulting in a high expression of the oncogene FOS Like 1, which contributes to glioblastoma growth, progression and poor prognosis [96].

A Turkish survey found a significantly higher expression of several TRP channels including TRPM7 in glioblastoma patients [97]. Notably, Tian et al. have reported that the pro-inflammatory prostaglandin E2, known to be elevated in glioblastoma, increases TRPM7 expression and responses as well as enhancing the migration and proliferation of tumor cells in human glioblastoma cells [98]. Moreover, TRPM7 in glioblastoma cells has been reported to promote the vesicular transfer of chloride intracellular channel 1 (CLIC1) from glioblastoma to endothelial cells. CLIC1 has been proposed to modulate the activity of neighboring endothelial cells promoting angiogenesis [99] (Table 1).

#### 4.1.2. Neuroblastoma

In a human neuroblastoma cell line, the MYCN oncogene has been reported to increase TRPM7 expression and thereby tumor cell proliferation and migration [100,101]. Interestingly, the inhibition of both TRPM7 channel and TRPM7 kinase activity by fingolimod, a disease-modifying immunmodulatory drug for the treatment of multiple sclerosis, has been observed in a human neuroblastoma MYCN-2 cell line. The concomitant changes in calcium signaling, loss of mitochondrial membrane potential and induction of apoptosis and autophagy may have clinical relevance, e.g., the sensitization of neuroblastoma cells to chemotherapeutic drugs. The possible repurposing of fingolimod for neuroblastoma was suggested [102].

In mouse neuroblastoma cells, TRPM7 was enriched at invadosomes (mechanosensory modules involved in cell adhesion and migration), and TRPM7 activity has been reported to regulate invadosome dynamics by affecting the tension–relaxation balance of the actomyosin cytoskeleton. This adhesion modulatory effect is supposed to be independent of Ca^2+^ influx through the channel [103,104,105]. A subsequent study revealed that TRPM7 enhanced metastatic potential, but not the proliferation rate in mouse neuroblastoma cells. TRPM7 overexpression conferred a malignant phenotype onto a poorly metastatic cell type in vivo. By maintaining progenitor-like features and controlling a developmental transcriptional program involving the transcription factor Snail family transcriptional repressor 2 (SNAI2), TRMP7 has been proposed to contribute to neuroblastoma progression [106]. TRPM7, as a part of a cytoskeletal complex, has been suggested to be involved in the control of dynamic formation and the function of cell adhesions and cellular protrusions in mouse neuroblastoma cell line. Components of this TRPM7 interactome correlated with human neuroblastoma metastasis and disease outcome [107]. Table 2 summarizes the role of the TRPM7 chanzyme in the pathophysiology of neuroblastoma as well as exemplifying the proliferative role of TRPM7 in retinoblastoma, another primitive childhood ectodermal tumor.

### 4.2. Head and Neck Cancers

#### 4.2.1. Nasopharyngeal Carcinoma

A higher expression of TRPM7 proteins was found in a human nasopharyngeal carcinoma cell line exhibiting metastatic ability compared to cells without metastatic potential. The overexpression of TRPM7 enhanced Ca^2+^ influx, which was critical for cell migration [109]. Studies on tissue samples derived from patients with nasopharyngeal carcinoma showed that TRPM7 overexpression correlated with tumor metastasis and predicted poor prognosis (worse survival). TRPM7 has been proposed to be critical in the migratory and invasive activity of tumor cells and metastasis. The downregulation of TRPM7 reduced the invasiveness and motility of the tumor cells [110]. Tumor proliferation regulated by TRPM7 was accompanied by the persistent activation of the JAK2/STAT3 signaling pathway. Moreover, the low expression of TRPM7 correlated with the better survival of patients with nasopharyngeal carcinoma. Remarkably, TRPM7 knockdown enhanced the sensitivity to irradiation [111].

#### 4.2.2. Laryngeal and Hypopharingeal Carcinomas

Functional expression of TRPM7 channels has been demonstrated in a human hypopharingeal squamous cell carcinoma cell line. The activation of these channels was critical for the growth and proliferation of these tumor cells [112]. The upregulation of TRPM7 by circular RNAs has been reported to enhance laryngeal squamous cell carcinoma progression [113]. A recent study has confirmed the high expression of TRPM7 in head and neck squamous carcinoma cells, especially in invasive cancer tissues. Importantly, TRPM7 expression correlated with worse survival. Overexpression was demonstrated in cisplatin-resistant cases compared with cisplatin-sensitive subjects. The silencing of TRPM7 suppressed several oncogenic signaling axes and reduced the migration, invasion, colony formation and tumorsphere formation of human squamous cell carcinoma cells in culture [114] (Table 3).

### 4.3. Malignant Neoplasms of the Upper Gastrointestinal Tract

#### 4.3.1. Esophageal Carcinoma

TRPM7 protein expression was only detected in esophageal squamous cell carcinoma cells but not in non-cancerous epithelia. However, surprisingly, TRPM7 was suggested to reduce the cell proliferation, migration and invasion of tumor cells [115].

#### 4.3.2. Gastric Cancer

The role of the TRPM7 channel in the pathophysiology of gastric cancer has been repeatedly demonstrated. Human gastric adenocarcinoma cells have been shown to express TRPM7 channels that are possibly involved in the growth and survival of tumor cells [55,116,117,118,119]. A ginsenoside and some plant extracts from Korea and China have been reported to induce apoptosis, inhibit cell growth and survival by blocking TRPM7 proteins in human gastric adenocarcinoma cells [120,121,122]. 5-lipoxygenase inhibitors, waixenicin A and quercetin have been also demonstrated to decrease the TRPM7 channel function and cause the death of gastric cancer cells [123,124]. A recent study has revealed that high TRPM7 expression is closely related to aggressive tumor behavior and an advanced stage, and was an indicator of poor prognosis in human gastric cancer [125] (Table 4).

#### 4.3.3. Pancreatic Cancer

The overexpression of TRPM7 has been repeatedly demonstrated in human pancreatic adenocarcinoma, enhancing cell proliferation, migration and invasion [76,127,128,129]. Aberrant TRPM7-mediated signaling via the Mg^2+^-sensitive Socs3a pathway was implicated in the uncontrolled proliferation of neoplastic cells and pancreatic carcinogenesis [127]. TRPM7 channels are expressed in pancreatic stellate cells involved in the development and progression of pancreatic ductal adenocarcinoma. The activation status of these cells seems to be correlated with TRPM7 expression. Furthermore, the overexpression of TRPM7 was also observed in cancer-associated fibroblasts [130]. A recent study revealed that TRPM7 regulates pancreatic stellate cell proliferation through the modulation of cell cycle regulators. TRPM7 silencing or pharmacological blockade resulted in the accumulation of cells in the G0/G1 phase (accompanied by an increase in p53 expression and reduced cyclin E, cyclin-dependent kinase 2 and proliferating cell nuclear antigen expression), and abolished cell growth as well as decreasing cell viability [130]. Moreover, the silencing of TRPM7 also enhanced the cytotoxic effect of gemcitabine [131].

TRPM7 has been suggested to increase cell migration through a Mg^2+^-dependent mechanism in human pancreatic cancer cells. TRPM7 expression correlated positively to a poorly differentiated status and reduced patient survival in pancreatic ductal adenocarcinoma [76]. Lefebvre et al. have reported that elastin-derived peptides known to stimulate cancer cell migration by interacting with their receptor, ribosomal protein SA (RPSA), stimulated TRPM7 currents in pancreatic cancer cells in culture. Furthermore, TRPM7 and RPSA co-localized at the plasma membrane in human pancreatic cancer cells, and this complex was proposed to regulate cell migration [132]. The control of the Hsp90α/urokinase/matrix metalloproteinase (MMP)-2 proteolytic axis was proposed as an underlying mechanism in the TRPM7-induced potentiation of pancreatic cancer cell invasion. Mg^2+^-entry through TRPM7 channels correlated with cell invasion and MMP secretion [77].

#### 4.3.4. Liver Cancer

An inverse relationship was observed between the TRPM7 channel activity and differentiation status of rat embryonic hepatocytes. Higher TRPM7 protein expression has been reported in proliferating rat hepatocytes compared with that in differentiated and noninvading cells in culture [133]. More recently, bradykinin has been demonstrated to promote the migration and invasion of human hepatocellular carcinoma cells by upregulating the expression of TRPM7 and facilitating the secretion of matrix metalloproteinase 2 (MMP2). The concomitant phosphorylation of non-muscle myosin heavy chain IIa and activation of calpains have been implicated in the relaxation of the actomyosin cytoskeleton and the turnover of peripheral adhesion complexes [134]. A pharmacological blockade of TRMP7 resulted in the oncogene-induced senescence of human hepatocellular carcinoma cells in vitro and in vivo. Importantly, it has been reported that TRPM7 proteins regulate RhoA activity, actin polymerization, subsequent Myocardin-related transcription factor A (MTRF-A)-Filamin A complex formation and MTRF-A/Serum Response Factor target gene expression. Both Mg^2+^ influx through the TRPM channel and the phosphorylation of RhoA by TRPM7 kinase have been suggested to be involved in the observed effect [38] (Table 5).

### 4.4. Colorectal Carcinoma

TRPM7 expression has been demonstrated in a colon cancer LoVo cell line. Remarkably, expression was lower in cells resistant to doxorubicin compared with doxorubicin-sensitive tumor cells, and TRPM7 was proposed to be involved in modulating drug resistance. High extracellular Mg^2+^ and consequently increased intracellular Mg^2+^ was suggested to downregulate TRPM7 through the activation of calpains in resistant cells. Lower TRPM7 expression also explained the slower proliferation of resistant cells [135,136]. Huang et al. have reported that TRPM7 drives human colon cancer cell proliferation, and this effect was independent of systemic Mg^2+^ status [137]. TRPM7 expression was higher in human colon tumor samples compared with that in non-neoplastic tissue, and adenocarcinomas showed a higher TRPM7 expression than did adenomas. Moreover, TRPM7 expression positively correlated with tumor grade [138]. An analysis of a public genomic database and patient data from a Chinese medical center revealed that expression of TRPM7 positively correlated with tumor infiltration, lymph node metastasis, distant metastasis and the clinical stage of colorectal cancer. The suppression of TRPM7 function reduced the cell proliferation, migration and invasion of colorectal cancer cells in vitro, by modulating EMT [139]. A recent analysis of public data revealed that TRPM7 expression was higher in younger patients with rectal cancer, and TRPM7 expression positively correlated with APC and KRAS gene expression, while it was inversely related to p53 expression in rectal cancer. Nevertheless, survival analysis showed that TRPM7 expression did not have any prognostic value in colon and rectal cancers [140].

The calcium/magnesium intake ratio and their balance regulated by parathyroid hormone and TRPM7 have been implicated in the development of colorectal carcinoma [141,142]. Furthermore, magnesium deficiency with physiological Ca^2+^ concentrations (i.e., increased Ca^2+^/Mg^2+^ ratio) was associated with increased TRPM7 expression, oxidative stress, induced calpain activity, increased cell migration and a more aggressive, metastatic phenotype of colon cancer cells [143] (Table 6).

### 4.5. Lung Cancer

The functional expression of TRPM7 receptors was demonstrated in a human non-small cell lung cancer cell line. EGF was shown to increase TRPM7 expression and function, and TRPM7 was suggested to be critically involved in the EGF-induced migration of tumor cells [144]. Correspondingly, TRPM7 was aberrantly expressed in human lung cancer tissue samples and various cell lines and was found to be an independent indicator of poor prognosis in lung cancer. Increased TRPM7 expression was associated with enhanced cancer stem-like and metastatic phenotypes. TRPM7 enhanced the Hsp90α/urokinase-type plasminogen activator (uPA)/MMP2 signaling pathway. TRPM7 silencing inhibited epithelial-to-mesenchymal transition, and suppressed stemness markers and phenotypes. Waixenicin A, a TRPM7 inhibitor, also suppressed the cancer stem cell phenotype of lung cancer cells [145]. More recently, the TRPM7/O-linked-β-N-acetyl glucosaminylation (O-GlcNAcylation) axis was suggested to represent a novel target for lung cancer therapy. Non-small cell lung cancer cell motility was shown to be modulated by TRPM7 and Ca^2+^-influx through O-GlcNAcylation. The hyper-O-GlcNAcylation of Caveolin-1 (Cav-1) and c-Myc resulted in increased tumor cell motility, and enhanced cell migration and invasion. Increased TRPM7 expression correlated with a reduced survival rate of tumor patients and more aggressive tumor phenotype. The inhibition of TRPM7 decreased cell motility and suppressed experimental lung metastases [146] (please see the summary of these data in Table 7).

### 4.6. Neoplasms of the Urinary System

#### 4.6.1. Kidney Cancer

An overexpression of TRPM7 in human renal cell carcinoma cell lines and tissues has been reported. It was associated with the increased proliferation and colony formation of tumor cells in vitro and tumor growth in vivo, in line with an aggressive phenotype and poor survival. TRPM7 has been shown to promote Akt phosphorylation, resulting in the repression of the tumor suppressor Forkhead box-O1 (FOXO1)’s transcriptional activity [147]. A prominent role of TRPM7 channels in the migration and invasion of human renal cell carcinoma cells via the Akt and Src pathways has been also suggested. The downregulation of TRPM7 prevented the migration and invasion of tumor cells [148] (Table 7).

#### 4.6.2. Bladder Cancer

Increased TRPM7 expression was observed in mouse MBT-2 bladder tumor cells compared with that in normal mouse urothelial cells [149]. TRPM7 was also overexpressed in human bladder cancer tissue and this increased expression was associated with tumor recurrence, metastasis and poor prognosis. TRPM7 knockdown by siRNA revealed that TRPM7 promoted bladder cancer cell proliferation, migration and invasion [150,151]. Cao et al. have reported increased TRPM7 expression and the dysregulation of proteins involved in EMT in human bladder cancer tissues and cancer cell lines. The downregulation of TRPM7 reversed the EMT status, inhibited the proliferation, migration and invasion of cancer cells, and promoted cell cycle arrest at the G0/G1 phase and apoptosis [152]. TRPM7 has been reported to regulate tumor growth, migration and invasion via the Src, Akt and c-Jun N-terminal kinase (JNK) signaling pathway in human bladder cancer cell lines [153]. In line with these data, oridonin, an anticancer compound, exhibited antiproliferative and antimigratory effects and induced apoptosis by suppressing TRPM7 expression and extracellular signal-regulated kinase (Erk) and Akt signaling in human bladder cancer cells in culture [154] (Table 7).

**Table 7 ijms-25-00719-t007:** Role of the TRPM7 chanzyme in the pathophysiology of lung cancer and of the neoplasms of the urinary tract.

Tumor Type/Experimental Sample	TRPM7 Expression	Channel/Kinase Function Involved	Observed/Implicated Signaling Mechanism	Major Finding	Ref.
Lung cancer					
human lung cancer cell line A549 and lung adenocarcinoma tissue	highly expressed both in cancer cells and human lung cancer tissue (no comparison with normal cells)	not elucidated	not investigated	TRPM7 was involved in EGF-induced migration of tumor cells	[144]
human lung cancer cell lines lung cancer tissue samples, Gene Expression Omnibus (GEO) dataset	overexpressed in human lung cancer tissue samples and cancer cell lines (compared to the non-tumor tissues and cells)	not elucidated	Hsp90α/uPA/MMP2 signaling pathway	TRPM7 overexpression enhanced the cancer stem cell-like and metastatic phenotypes. TRPM7 was an independent indicator of poor prognosis	[145]
lung carcinoma tissues from Bittner’s dataset in Oncomine database, lung cancer cell lines, xenograft mouse model	expression was higher in carcinoma patients with poorer prognosis	dependent on Ca^2+^ influx	Hyper-O-GlcNAcylation of Cav-1 and c-Myc	TRPM7 activation was linked to increased tumor cell motility, enhanced cell migration and invasion. Inhibition of TRPM7 suppressed experimental metastases. TRPM7 overexpression was associated with poorer prognosis	[146]
Neoplasms of the urinary tract					
human renal cancer tissue, renal cancer cell lines, xenograft mouse model	overexpression in renal cancer cells and tissues compared with normal cells or non-neoplastic tissues	not discussed	PI3K/Akt/FOXO1 pathway	TRPM7 was associated with increased proliferation and colony formation of tumor cells in vitro and tumor growth in vivo, in line with an aggressive phenotype and poor survival	[147]
human renal cancer cell lines	expression was confirmed and then manipulated	not discussed	Src/Akt signaling pathway	prominent role of TRPM7 in migration and invasion of human renal cell carcinoma cells	[148]
mouse bladder tumor MBT-2 cells, human bladder carcinoma cell line T24, primary mouse urothelial cells	increased expression in mouse bladder tumor cells compared with normal cells	not discussed	not investigated	increased TRPM7 expression in tumor cells	[149]
human bladder cancer tissue	overexpressed in bladder cancer patients compared with controls	not discussed	not investigated	increased TRPM7 expression in cancer tissue	[151]
human bladder cancer cell lines and bladder cancer tissue samples	overexpressed in bladder cancer tissues compared with non-tumor bladder tissues	not discussed	not investigated	TRPM7 was overexpressed in bladder cancer and correlated with poor prognosis. TRPM7 was involved in cell proliferation, apoptosis, migration and invasion abilities of cancer cells	[150]
human bladder cancer cell lines and bladder cancer tissue samples	TRPM7 was overexpressed in bladder cancer tissues compared with the normal bladder tissues	not discussed	MAPK signaling pathway	suppression of TRPM7 inhibited proliferation, migration and invasion of cancer cells, and promoted cell cycle arrest at G0/G1 phase and apoptosis	[152]
human bladder cancer cell lines, mouse xenograft model	expression was confirmed and then manipulated	not discussed	Src, Akt, and JNK pathways	TRPM7 promoted tumor growth, migration and invasion	[153]
human bladder cancer cell line T24; mouse xenograft model	expression was confirmed and then reduced by oridonin	not discussed	Erk and Akt signaling was implicated	oridonin exhibited anti-proliferative and anti-migratory effects and induced apoptosis by suppressing TRPM7 expression	[154]

### 4.7. Breast Cancer

The oncogenic roles and therapeutic potentials of TRPM7 in breast cancer are intensely investigated. More details are available in a recent comprehensive review [49].

The functional expression of TRPM7 and thre regulation of tumor cell proliferation by this channel have been reported in human breast cancer cells. TRPM7 was overexpressed in grade III tumors, and its expression positively correlated with the mitosis marker Ki67 [155]. Subsequently, a high expression of TRPM7 in human breast cancer cells and tissue, as well as its correlation with proliferation, has been repeatedly demonstrated [67,156,157,158]. Remarkably, in a Chinese population, an association between the single-nucleotide polymorphism of the TRPM7 gene and breast cancer has been observed [159]. Epigenetic alterations of TRPM7 are supposed to influence the clinical outcome of breast tumors. A recent study has demonstrated that TRPM7 was highly expressed in the luminal A subtype of breast cancer. Nevertheless, the promoter methylation of TRMP7 was associated with a better prognosis of the disease [160].

Interestingly, increased TRPM7 expression was found in human infiltrating ductal breast carcinoma samples with microcalcifications compared with that in tumors without calcification [161]. This mineralization is associated with poor prognosis and is suppressed by the inhibition of TRPM7 or the chelation of intracellular calcium in human breast cancer cells. Increased Mg^2+^ resulted in a protective effect, and this protection was independent of TRPM7 activity [162]. On the other hand, low serum Mg^2+^ activating TRMP7 has been suggested to increase intracellular Ca^2+^ and Ca^2+^-related cell proliferation. The resulting high serum Ca^2+^/Mg^2+^ ratio has been hypothesized to be a risk factor for postmenopausal breast cancer [163].

Recent studies showed that TRPM7 inhibitors interrupted the cell cycle, arresting the cells in the S phase, which suppresses the viability and migration of TRPM7 expressing breast cancer cells. Another investigation indicated that the TRMP7 antagonist carvacrol increased the proportion of cells in the G1/G0 phase and decreased the proportion of cells in the S and G2/M phases by regulating cyclin proteins. As a result, the suppression of cell growth and proliferation has been observed in human breast cancer cells [164,165,166]. TRPM7 activity was shown to be associated with metastatic potential and migratory properties in human breast cancer tissue and cells. Furthermore, TRPM7 is part of the mechanosensory machinery that regulates cellular tension and steers adhesion dynamics thereby, promoting cell migration and metastasis formation [156]. TRPM7 kinase domain and not Ca^2+^ influx through the channel was suggested to be involved in breast cell migration via the phosphorylation of myosin IIA in metastatic estrogen receptor (ER) breast cancer. TRPM7 was implicated in the invasiveness and dissemination of tumor cells [157]. In line with this observation, the blockade of TRPM7 kinase suppressed breast cancer cell migration and invasion in human breast cancer cell lines [167].

On the other hand, the activation of MAP kinase pathway via increased intracellular calcium levels in response to TRPM7 activation has also been proposed to cause an enhanced migration and invasion capacity of a human breast cancer cells in culture [158]. TRPM7 was proposed to be a partial regulator of EMT in breast cancer cells [168]. Davis et al. have reported that the induction of EMT in breast cancer cells in response to TRPM7 activation depends on a cytosolic calcium signal. These mechanisms involve the regulation of EGF-induced STAT3 phosphorylation and vimentin expression. Another study also reported the partial involvement of TRPM7 in EGF-induced vimentin protein expression, but its calcium-dependent regulation was shown to occur via a TRPM7-independent pathway [169]. Wang et al. raised the possibility that the overexpression of TRPM7 in breast cancer cells resulted in an increased intracellular zinc concentration, which increased the level of MDMX (a negative regulator of the tumor suppressor p53), thereby promoting cell migration. The zinc-permeable channel function but not the kinase domain of TRPM7 was suggested to play an essential role in this observation [170]. TRPM7 inhibitors have been reported to synergistically interact with tumor necrosis factor-related apoptosis-inducing ligand (TRAIL) to enhance antiproliferative effects and apoptosis in triple-negative breast cancer cells. This effect was thought to be involved in TRPM7 channel activities (reduced Ca^2+^ influx and the downregulation of cellular FLICE inhibitory protein) rather than TRPM7 kinase activities [171].

The expression of the transcription factor Sex-determining region Y-related high-mobility group-BOX gene 4 (SOX4), a regulator of EMT in breast cancer cells, was promoted by TRPM7 via mechanical regulation by reducing myosin II-based cellular tension. This led to cytoskeletal relaxation and drove the mesenchymal features of breast tumor cells. TRPM7 has been also considered part of a mechanical signaling hub that controls cell plasticity [172] (Table 8).

### 4.8. Malignant Tumors of the Female Reproductive Organs

#### 4.8.1. Ovarian Cancer

TRPM7 has been reported to be highly expressed in ovarian cancer and significantly associated with poor prognosis, tumor progression and decreased disease-free survival in a Chinese population [173]. A subsequent study has revealed that TRPM7 mediates cell proliferation, migration and invasion in human ovarian cancer cell lines at least in part by influencing Akt, Src and p38 signaling pathways and the formation of cell adhesion complexes [174]. TRPM7 expression correlates negatively with E-cadherin, but positively with N-cadherin, vimentin and Twist expression in human ovarian cancer cells. The upregulation of TRPM7 expression is associated with increased EMT. TRPM7 silencing decreased the EGF-induced migration, invasion and wound healing of cancer cells in vitro and metastasis formation in mice in vivo. The attenuation of calcium-related PI3K/Akt activation has been suggested as an underlying mechanism [175]. Chen et al. have proposed that TRPM7 modulates glucose metabolic reprogramming involved in ovarian tumor cell proliferation. TRPM7 silencing suppressed the proliferation of ovarian cancer cells by shifting glycolysis to oxidative phosphorylation. This event was accompanied by decreased glucose uptake, increased AMP-activated protein kinase (AMPK) phosphorylation and decreased hypoxia-inducible factor (HIF)-1α protein levels in ovarian cancer cells. TRPM7 levels were negatively correlated with IDH3B and UQCRC1, but correlated positively with hexokinase 2 (HK2) and pyruvate dehydrogenase kinase (PDK)1 expression in ovarian cancer tissue [176] (Table 9).

#### 4.8.2. Uterine Tumors

Cervical cancer cell proliferation, invasion and migration were suppressed by miR-543 in vitro and tumor growth in vivo in part targeting TRPM7. PI3K/Akt and p38/MAPK pathways have been suggested to be involved in miR-543/TRPM7-regulated cervical cancer development [177]. Furthermore, tumor suppressor miR-192-5p inhibited cervical cancer cell proliferation and invasion by targeting TRPM7 [178]. Progesterone has been shown to reduce TRPM7 expression in human cervical cancer cells reducing cell proliferation and switching from necrosis to apoptosis [179].

A Turkish retrospective study revealed that TRPM7 may be a progression marker for endometrial hyperplasia (regardless of if it was categorized as “with atypia” or “without atypia”) [180] (Table 9).

**Table 9 ijms-25-00719-t009:** Role of the TRPM7 chanzyme in the pathophysiology of ovarian and uterine cancer.

Tumor Type/Experimental Sample	TRPM7 Expression	Channel/Kinase Function Involved	Observed/Implicated Signaling Mechanism	Major Finding	Ref.
human ovarian cancer tissue samples	increased expression in ovarian carcinomas compared with normal tissues	not investigated	not investigated	increased TRPM7 expression correlated with tumor progression and poor prognosis	[173]
human ovarian cancer cell lines	increased expression in ovarian cancer cells compared with normal cells	TRPM7 mediated Ca^2+^ influx was suggested	Akt, Src and p38 signaling pathways	TRPM7 mediated cell proliferation, migration and invasion in cancer cells	[174]
human ovarian cancer tissue samples and cancer cell lines, mouse xenograft model	overexpressed in tumor tissues compared with non-tumor tissues	TRPM7 mediated Ca^2+^ influx was suggested	PI3K/Akt pathway	TRPM7 was involved in the EMT, migration, invasion and metastasis of uterine cancer cells	[175]
human ovarian cancer tissue samples and cancer cell lines, mouse xenograft model	expression was confirmed and then manipulated	not discussed	TRPM7 silencing/AMPK/HIF-1α axis	TRPM7-modulated glucose metabolic reprogramming was involved in ovarian tumor cell proliferation	[176]
human cervical cancer cell lines and normal human cervical epithelial cell line; cervical cancer tissue samples; mouse xenograft model	expression was manipulated	not discussed	PI3K/Akt and p38/MAPK pathways	TRPM7 was involved in proliferation, invasion and migration of cancer cells	[177]
human cervical cancer cell lines, mouse xenograft model	expression was manipulated	not discussed	MAPK was indirectly implicated	TRPM7 was involved in cervical cancer cell proliferation, migration and invasion	[178]
human cervical cancer cell line HeLa and human embryonic kidney cells	expression was manipulated	not discussed	not investigated	reduced TRPM7 expression reduced cell proliferation and resulted in switching from necrosis to apoptosis	[179]
human endometrium tissue samples	in early-stage endometrial, cancer reduction in TRPM7 expression was relative to endometrial hyperplasia	not discussed	not investigated	TRPM7 was suggested as a progression marker for endometrial hyperplasia (regardless of the atypical criteria)	[180]

### 4.9. Prostate Cancer

The increased expression of TRPM7 in human prostate cancer cell lines has been repeatedly published. Moreover, a higher Ca^2+^/Mg^2+^ ratio in prostate cancer patients resulted in increased Ca^2+^ influx mediated by TRPM7, which promoted cell proliferation [181]. TRPM7 was suggested to be involved in cholesterol-induced cell proliferation and the migration of human prostate cells. Cholesterol increased Ca^2+^ entry via the TRPM7 channel, promoting the proliferation of prostate cells via the activation of the Akt and/or Erk pathway. Increased calpain activity and decreased E-cadherin expression was proposed to facilitate the migration of tumor cells [182].

The accumulation of HIF-1, playing a central role in tumor progression, is regulated by TRPM7 through a non-conventional RelB-dependent nuclear factor kappa B (NFκB) signaling pathway and the regulation of superoxide activity in osteosarcoma and prostate cancer cell lines [183]. The anti-apoptotic effect of TRPM7 has been demonstrated in prostate cancer cells. TRPM7 inhibitors enhanced TRAIL-induced apoptosis in these cells [184]. The TRPM7 antagonist carvacrol reduced prostate cancer cell proliferation, migration and invasion in prostate cancer cell lines. Reduced MMP-2 protein expression and F-actin dynamics as well as the suppression of PI3K/Akt and MAP kinase pathways have been shown to be involved in this effect [185]. Chen et al. have also reported the suppressed migration and invasion of prostate cancer cells in response to the downregulation of TRPM7. The influence on the EMT (increased E-cadherin and Paxilin while decreased MMP-2 and MMP-9) was the underlying mechanism. TRPM7 seems to have a key role in metastasis [74]. Transforming growth factor beta (TGFβ), known to promote the metastasis of prostate cancer inducing cell migration, and enhanced TRPM7 expression and function in prostate cancer cells. The consecutive Mg^2+^ influx was essential for EMT-induced cell migration [186]. A recent study revealed the mechanism of the involvement of TRPM7 in the growth and metastatic ability of prostate cancer cells under hypoxic conditions. Hypoxia increased TRPM7 expression and HIF-1α accumulation in androgen-independent prostate cancer cells. Annexin A1 has been suggested as a downstream mediator of EMT. TRPM7 knockdown promoted oxygen-independent RACK1-mediated HIF-1α degradation, resulting in the suppression of EMT, cell migration and invasion [187] (please see the summary of these data in Table 10).

### 4.10. Other Neoplastic Pathologies

#### 4.10.1. Melanoma

The expression of TRPM7 was reported in melanocytes and in cell lines derived from melanoma [188]. Guo et al. have hypothesized that TRPM7 is a marker of deregulated metabolic activity in melanoma cells as well as a target for the restoration of protective and detoxifying properties in normal melanocytes [78]. A recent study revealed that TRPM7 was involved in the proliferation of canine and human non-UV-induced melanomas. TRPM7 genes were suggested as candidate oncogenes for human mucosal melanoma [189] (Table 10).

#### 4.10.2. Multiple Myeloma

TRPM7 has been shown to be upregulated in myeloma cells compared with normal plasma cells. The knockdown or inhibition of TRPM7 inhibited myeloma cell migration and dissemination [190,191]. Myeloma cell motility and dissemination were suggested to be modulated by the Ca^2+^ influx-O-GlcNAcylation regulatory axis linked to integrin α4 and β7. The inhibition of Ca^2+^ influx channels such as TRPM7 repressed the aggressive tumor phenotype [191] (Table 10).

**Table 10 ijms-25-00719-t010:** Role of the TRPM7 chanzyme in the pathophysiology of prostate cancer, melanoma and multiple myeloma.

Tumor Type/Experimental Sample	TRPM7 Expression	Channel/Kinase Function Involved	Observed/Implicated Signaling Mechanism	Major Finding	Ref.
Prostate cancer					
human prostate cancer cell lines and prostate cancer tissue samples	expression was increased in cancer cells and tissue samples when compared with control	Ca^2+^ influx was suggested	not investigated	TRPM7 activity was involved in cell proliferation	[181]
human prostate cancer cell lines and prostate cancer tissue samples	expression was confirmed, cholesterol increased it in both cancer and healthy tissues	Ca^2+^ influx was suggested	Akt and/or Erk pathway	TRPM7 was involved in cell proliferation and migration	[182]
human prostate, breast, colorectal, osteosarcoma and liver cancer cell lines:	not tested in prostate cancer cells	not discussed	NFκB signaling pathway	accumulation of HIF-1, playing a central role in tumor progression, was regulated by TRPM7 in cancer cells	[183]
human prostate cancer cell line PC3	expression was confirmed and then manipulated	not discussed	PI3K/Akt pathway.	TRPM7 had anti-apoptotic effect	[184]
human prostate cancer cell lines	expression was higher in prostate cancer cells than in control cells	not discussed	PI3K/Akt and MAP kinase pathways	TRPM7 was involved in prostate cancer cell proliferation, migration and invasion	[185]
human prostate cancer cell lines, prostate hyperplasia cells and normal prostate cells; human prostate tissue samples, Gene Expression Omnibus (GEO) database	expression was increased in prostate cancer cells compared with prostate hyperplasia and normal cells. TRPM7 was upregulated in metastatic prostate cancer tissues, compared with primary cancer tissues and benign hyperplasia tissues	calcium signaling, was not discussed in details	calcium signaling pathway was suggested, resulting in increased E-cadherin and Paxilin while decreased MMP-2 and MMP-9	TRPM7 was involved in migration and invasion of prostate cancer cells	[74]
human prostate cancer cell lines	expression was higher in cancer cells compared with normal cells	TRPM7-mediated Mg^2+^ influx was implicated	not investigated in detail	TRPM7 was involved in EMT-induced cell migration	[186]
human prostate cancer cell lines, Human Protein Atlas (HPA) database	high expression of TRPM7 gene in prostate cancer patients was associated with poor survival; hypoxia increased TRPM7 expression	Ca^2+^ entry was proposed	Ca^2+^ entry was suggested to activate the calcineurin phosphatase activity resulting in HIF-1α/Annexin A1 signaling	involvement of TRPM7 in growth and metastatic ability of prostate cancer cells under hypoxic conditions	[187]
Other neoplasms (melanoma, multiple myeloma)					
human melanoma cell lines, zebrafish trpm7 mutants	expression was higher in melanoma cells compared with that in normal melanocytes	not discussed	not fully elucidated	TRPM7 expression was higher in melanoma cells; melanophores required TRPM7 to detoxify intermediates of melanin synthesis	[188]
canine melanoma tissue samples, canine melanoma cell lines, human melanoma cell lines	TRPM7 was identified as candidate oncogene via genomic and transcriptomic analysis of canine melanoma samples	not discussed	not investigated	TRPM7 was involved in proliferation of canine and human non-UV-induced melanomas	[189]
human multiple myeloma derived cell lines; mouse xenograft model; Oncomine database	expression was higher in myeloma cells compared with that in normal plasma cells	Ca^2+^ influx was suggested	Ca^2+^ influx-O-GlcNAcylation regulatory axis	knockdown or inhibition of TRPM7 inhibited myeloma multiplex cell migration and dissemination	[191]

## 5. Conclusions

Here, we provided detailed descriptions of the role of TRPM7 in numerous tumor types of various organs and tissues, including expression profiles, tumor hallmarks and intracellular signaling pathways in an easy and quick-to-understand way with 10 tables. Data show that the TRPM7 chanzyme is expressed in most tumor cell types. In many cases, overexpression is detected, which is associated with an increase in tumorigenesis. However, the interpretation of these data requires caution from several points of view. In many studies, TRPM7 expression levels in tumor cells were not compared to thoser of normal cells. Furthermore, when its blockade might promise the inhibition of tumorigenesis, it is usually not clarified to what extent TRPM7 is a physiological part of normal functioning. Considering these uncertainties, how could we influence TRPM7 pathological functions selectively without affecting its physiological roles? This overview shows that in many cases, the signaling cascade involved in tumorigenesis was not clarified, including whether it is involved solely in aberrant/pathological or also in indispensable physiological functions. The TRPM7 chanzyme may be coupled to various signaling cascades in response to the influx of Ca^2+^, Mg^2+^ or other divalent cations via the channel or phosphorylation by its kinase. A better understanding of the peculiarity of its double-channel/kinase function might provide a basis for the separation of physiological and pathophysiological functions of the chanzyme. However, only few tumor-related studies aimed to further elucidate this aspect. Revealing the regulation ofTRPM7 ion conductance under pathological conditions warrants further studies in the future. In addition, an understanding of changes in the trafficking and homo- and hetero-oligomerization of the chanzyme under pathological circumstances requires detailed examinations as well.

It is noteworthy that some experimental data are not in agreement with the general view that TRPM7 is involved in tumorigenesis, suggesting that it may even have a protective effect. A possible explanation for this observation is that different species and organ systems were investigated, different experimental models were used and the test conditions were different. However, the peculiarities of the functioning of the subtly regulated channel/kinase system might provide a more thorough and precise explanation of these data. Its clarification is of enormous scientific and practical interest. A better understanding of the underlying mechanisms of the TRPM7 dysfunctions resulting in malignancies may open the avenue for the prevention and more effective treatment of malignant disorders, thereby saving lives, alleviating suffering and saving healthcare resources.

## Figures and Tables

**Figure 1 ijms-25-00719-f001:**
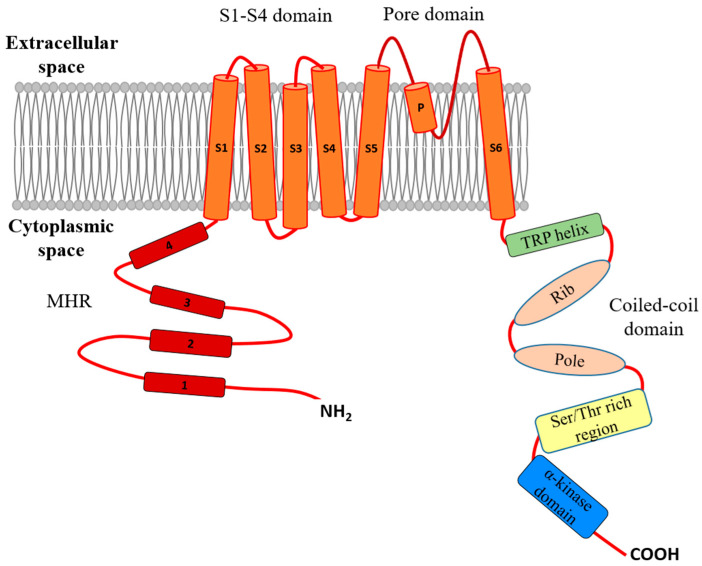
Schematic representation of the protein structure of the TRPM7 chanzyme.

**Figure 2 ijms-25-00719-f002:**
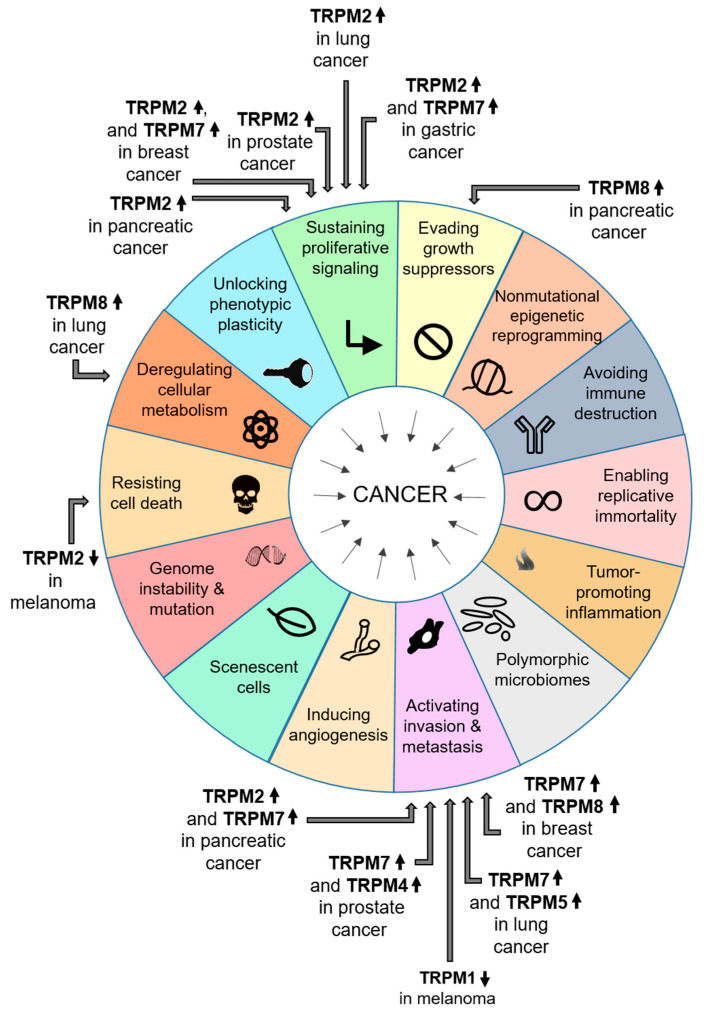
Role of individual TRPM channels in the hallmarks of the various types of cancer. ↑ indicates the increase and ↓ indicates the decrease in expression for each TRPM protein. For the references, please see [52,53,54,55,62,67,68,69,70,71,72,73,74,75,76,77,78,79,80,81,82].

**Table 1 ijms-25-00719-t001:** Role of TRPM7 chanzyme in the pathophysiology of glioblastoma.

Tumor Type/Experimental Sample	TRPM7 Expression	Channel/Kinase Function Involved	Observed/Implicated Signaling Mechanism	Major Finding	Ref.
human glioblastoma cell line U87	higher expression than that in normal human astrocytes	phosphorylation is implicated, but not discussed in detail	MMP-2 expression, phosphorylation of cofilin, Ras/MEK/MAPK and PI3K/Akt signaling	TRPM7 antagonism reduced the viability, migration and invasion of tumor cells	[87]
human glioblastoma cell line U251	expressed (was not compared with normal astrocytes)	both Ca^2+^ entry and phosphorylation are implicated but not discussed in detail	PI3K/Akt and MEK/ERK signaling	TRPM7 blockade reduced viability, proliferation, and migration of tumor cells	[86]
human glioblastoma cell line A172 and human glioma tissue samples	expression of functional TRPM7 channels (was not compared with normal astrocytes)	not discussed	not investigated	suppression or inhibition of TRPM7 reduced the proliferation, migration, and invasion of tumor cells	[84]
human malignant glioma MGR2 cells; Oncomine Database	expression was higher in human malignant glioma tissues compared with normal brain tissues	not discussed	influence on the expression of cell cycle regulators (checkpoint Rb, cyclins and their dependent kinases; cyclin inhibitory proteins)	inhibition of TRPM7 led to G0/G1 cell cycle arrest and cell proliferation loss	[85]
rat C6 glioma and human A172 glioblastoma cells	not investigated	not discussed	not investigated	inhibition of TRPM7 channels suppressed the proliferation of glioma cells	[88]
human glioma cell lines	expression in glioma cells was higher than in dental stem cells	not discussed	not investigated	TRPM7 was involved in ATP-induced inhibition of methuosis (non-apoptotic cell death of tumor cells)	[89]
human U87 glioblastoma cell line	not investigated	both sustained and prolonged Ca^2+^ influx were involved, involvement of kinase activity was also supposed	MMP-2 upregulation; MAPK/ERK signaling but not the PI3k/Akt pathway	TRPM7 activation enhanced U87 cell migration and invasion but not viability and proliferation	[91]
human glioblastoma cell lines; xenograft mouse model	expression confirmed	not discussed	PI3K/Akt related downstream effectors	TRPM7 inhibition reduced cell viability, migration and invasion	[90]
human glioblastoma cell lines; xenograft mouse model	expression confirmed	not discussed	PI3K/Akt/GSK3β signaling	TRPM7 antagonism reduced the glioblastoma tumor volume in xenograft mouse model	[92]
human glioblastoma A172 cell line, neurospheroid culture	expression confirmed	not discussed	JAK2/STAT3 and/or Notch signaling resulting in activation of ALDH1 promoters	TRPM7 promoted proliferation, migration and invasion of glioma cells	[93]
human glioma/glioblastoma cell lines, tumorsphere culture; Oncomine Database	increased expression in glioblastoma compared with normal brain tissues	functional coupling between TRPM7 channel and a kinase domain was necessary for TRPM7-related signaling	Notch1 signaling, expression of CD133 and ALDH1.	TRPM7 was responsible for regulation of glioma stemness, contributed to glioma cell growth and invasion	[94]
human glioma/glioblastoma cell lines	expression was manipulated	channel activity was required for glioma cell growth while the kinase domain was required for cell migration/invasion	negative regulation of the tumor suppressor microRNA(miR)-28-5p) increased Rap1b expression.	TRPM7 increased glioma cell proliferation and migration/invasion	[95]
various glioma and glioblastoma cell lines	expression was manipulated	involvement of both the TRPM7 channel and kinase domain was suggested	downregulation of miR-301a-3p resulting in high expression of the oncogene FOS Like 1	TRPM7/HOX transcript antisense intergenic RNA axis was overexpressed in glioma cells promoting cell proliferation and invasion.	[96]
human glioblastoma tissue samples	significantly higher expression compared to the control	not discussed	not investigated	higherTRPM7 expression in glioblastoma patients compared to the control	[97]
human glioblastoma A172 cells; human embryonic kidney cells	expressed in tumor cells; expression was increased in response to PGE2	not discussed	not investigated	PGE2 enhanced migration and proliferation of human glioblastoma cells via the upregulation of TRPM7	[98]
human primary microvascular HMEC and glioblastoma U87MG cells	expressed	not discussed	Ca^2+^ signaling (cytosolic Ca^2+^ waves)	TRPM7 was involved in the progression of glioblastoma by promoting the vesicular transfer of CLIC1 from glioblastoma to endothelial cells	[99]

**Table 2 ijms-25-00719-t002:** Role of the TRPM7 chanzyme in the pathophysiology of neuroblastoma.

Tumor Type/Experimental Sample	TRPM7 Expression	Channel/Kinase Function Involved	Observed/Implicated Signaling Mechanism	Major Finding	Ref.
human neuroblastoma cell lines and retinal pigment epithelial cells; human neuroblastoma tumor datasets: NB88 and Kocak-649	expressed in all tumor samples (was not compared with non-tumor cells), expression was higher in MYCN amplified tumors	not discussed	not investigated	MycN oncogene promoted TRPM7 expression and thereby tumor cell proliferation and migration. Survival probability was higher in patients with low TRPM7 expression	[101]
human neuroblastoma cell line SHEP-21N; human neuroblastoma tumor dataset Kocak-649	expression was found in all tumor samples (was not compared with non-tumor cells), and correlated with MYCN amplification	not discussed	not investigated	genetic suppression of TRPM6/TRPM7 inhibited cell proliferation	[100]
human neuroblastoma cell lines; human neuroblastoma tumor dataset Kocak-649	expression was found in tumor samples (was not compared with non-tumor cells)	both channel and kinase are required	myosin IIA and histone H3 phosphorylation	TRPM7 expression correlated with lower patient survival; inhibition of TRPM7 resulted in loss of mitochondrial membrane potential and induction of apoptosis and autophagy	[102]
N1E-115 mouse neuroblastoma cells	expressed, it was localized to cell adhesion structures	Ca^2+^- and kinase-dependency	inhibition of myosin II, potentially involving myosin IIA heavy chain phosphorylation	TRPM7 promoted relaxation of actomyosin cytoskeleton, thereby affecting cell adhesion	[104]
N1E-115 mouse neuroblastoma cells	overexpressed in cell line	independent of (localized) Ca^2+^ influx	further not specified	TRPM7 regulated invadosome dynamics by affecting the tension–relaxation balance of the actomyosin cytoskeleton	[105]
mouse and human neuroblastoma cell lines; mouse xenograft model	expression was manipulated	not discussed	TRPM7 controls a developmental transcriptional program involving the transcription factor SNAI2—details not specified	TRPM7 enhanced metastatic potential, but not proliferation rate, and contributed to tumor progression; TRPM7 expression closely associated with the migratory and metastatic properties	[106]
mouse neuroblastoma cells; human neuroblastoma tumor datasets	coexpressed with interacting proteins involved in cell adhesion and metastasis	not discussed	not investigated	expression of TRPM7-interacting proteins correlated with neuroblastoma progression	[107]
human retinoblastoma cells, human embryonic kidney cells	abundant expression (was not compared with non-tumor cells)	Ca^2+^ influx was related to proliferation of cells	Ca^2+^ influx dependent mechanism—not specified further	TRPM7 promoted cell growth of tumor cells	[108]

**Table 3 ijms-25-00719-t003:** Role of the TRPM7 chanzyme in the pathophysiology of head and neck cancers.

Tumor Type/Experimental Sample	TRPM7 Expression	Channel/Kinase Function Involved	Observed/Implicated Signaling Mechanism	Major Finding	Ref.
human nasopharyngeal carcinoma SUNE1 cells	higher expression in cells with metastatic ability compared to cells without metastatic potential	not discussed (only channel function was investigated)	effect was dependent both on influx of Ca^2+^ and release of Ca^2+^ from intracellular stores via RyRs, but not IP3Rs	overexpression of TRPM7 enhanced cell migration	[109]
human nasopharyngeal carcinoma cell lines, and normal nasopharyngeal epithelial cell lines; tissue samples from nasopharyngeal carcinoma patients	highly expressed in tumor cells and tissues (low expression in normal epithelial cells); overexpression associated with metastasis	not discussed	not investigated	TRPM7 overexpression correlated with tumor metastasis and predicted poor prognosis	[110]
human nasopharyngeal carcinoma cell lines; nasopharyngeal tumor tissue	expression was upregulated in tumor tissues compared to nearly negative expression in normal mucosa	not discussed	JAK2/STAT3 signaling and downstream proteins of STAT3	TRPM7 promoted tumor proliferation; its low expression correlated with better survival	[111]
human hypopharyngeal carcinoma FaDu cell line	expression of TRPM7 (was not compared with normal cells)	not discussed	increased intracellular Ca^2+^	TRPM7 was critical for the growth and proliferation of tumor cells	[112]
laryngeal carcinoma tissue samples; xenograft mouse model	expression was manipulated	not discussed	not investigated	TRPM7 was involved in proliferative and invasive ability of tumor cells	[113]
human head and neck carcinoma tissue samples, squamous cell carcinoma cell lines, oral normal keratinocytes; xenograft mouse model; Cancer Genome Atlas (TCGA) dataset, GSE26549 dataset	expression was higher in tumor samples than in normal tissues; it was especially high in invasive cancer tissues	not discussed	relationship between TRPM7 and the calcineurin/NFAT pathway (NFATC3, NOTCH1)	higher TRPM7 expression correlated with worse survival; silencing of TRPM7 reduced migration, invasion, colony formation and tumorsphere formation of tumor cells in culture, and suppressed metastasis in tumor xenograft model	[114]

**Table 4 ijms-25-00719-t004:** Role of the TRPM7 chanzyme in the pathophysiology of esophageal and gastric cancer.

Tumor Type/Experimental Sample	TRPM7 Expression	Channel/Kinase Function Involved	Observed/Implicated Signaling Mechanism	Major Finding	Ref.
human esophageal tumor samples and cell lines	expression was detected in esophageal squamous cell carcinoma cells but not in non-cancerous epithelia	not discussed	not investigated	survival rate of patients with high TRPM7 expression was higher than that of patients with low expression	[115]
human gastric adenocarcinoma cell lines	expressed in human gastric adenocarcinoma cell lines (was not compared with normal cells)	not discussed (only channel function was investigated)	role in Mg^2+^ homeostasis (Mg^2+^ was critical for the growth and survival of tumor cells)	TRPM7 channels were involved in tumor cell growth and survival	[116]
human gastric adenocarcinoma cell line (AGS); human embryonic kidney (HEK293) cells	expression was manipulated	not discussed	if investigated, MAPK was implicated	blockade or suppression of TRPM7 induced apoptosis, and inhibited cell growth and survival	[117,118,120,121,122,123,124,126]
human gastric carcinoma tissue samples	highly expressed in gastric cancer tissues compared to noncancerous tissues	not discussed	not investigated	high TRPM7 expression was closely related to aggressive tumor behavior and an advanced stage, and was an indicator of poor prognosis	[125]

**Table 5 ijms-25-00719-t005:** Role of the TRPM7 chanzyme in the pathophysiology of pancreatic and liver cancer.

Tumor Type/Experimental Sample	TRPM7 Expression	Channel/Kinase Function Involved	Observed/Implicated Signaling Mechanism	Major Finding	Ref.
human pancreatic adenocarcinoma tissues and cell lines	was overexpressed in cancer tissue compared with normal pancreatic tissue	was not discussed in view of tumorigenesis	“aberrant” Mg^2+^-sensitive Socs3a pathway	TRPM7 was overexpressed and was involved in proliferation in pancreatic adenocarcinoma	[127]
human pancreatic adenocarcinoma tissues and cell line	was overexpressed in cancer tissues; TRPM7 staining was stronger in tumors of a high grade	not discussed	Mg^2+^-dependent mechanism (was not further specified)	TRPM7 expression correlated with progression of the tumors and was inversely related to patient survival. TRPM7 increased cell migration.	[76]
human pancreatic adenocarcinoma tissue and cell lines	was overexpressed in a proportion of the pre-malignant lesions and malignant tumors of the pancreas (compared with normal tissue)	not discussed	not investigated	TRPM7 was overexpressed in pancreas carcinoma, and was necessary for pancreatic cancer cell invasion	[129]
human pancreatic stellate cell lines; pancreatic cancer PAAD dataset	expression correlated with the activation status of pancreatic cells, TRPM7 was overexpressed in cancer-associated fibroblasts	cation entry was proposed as an underlying mechanism	Mg^2+^-dependent PI3K/Akt but not ERK pathway, affecting p53 and allowing the G1-S transition	TRPM7 expression correlated with the activation status of pancreatic cells and regulated their proliferation	[130]
human pancreatic adenocarcinoma cell lines	expression was manipulated	not discussed	not elucidated	TRPM7 was required for preventing non-apoptotic cell death through replicative senescence; silencing of TRPM7 enhanced cytotoxic effect of gemcitabine	[131]
human pancreatic cancer cell line MIA PaCa-2	TRPM7 and ribosomal protein SA receptors were co-localized in cancer cells	not discussed	modification of monovalent currents through TRPM7 was suggested but was not further elucidated	cancer cell migration was TRPM7-dependent	[132]
human pancreatic cancer cell lines and pancreatic adenocarcinoma samples	expression was confirmed in tumor cells; expression in metastasis correlated with expression levels in primary tumor	Mg^2+^ entry and possibly kinase activation were proposed	regulation of Hsp90α/uPA/MMP-2 proteolytic pathway	TRPM7 potentiated pancreatic cancer cell invasion	[77]
rat hepatoma WIF-B cells, HEK293T cells	overexpressed in proliferating cells compared with differentiated and non-dividing rat hepatocytes	not discussed	not investigated	TRPM7 was more highly expressed in proliferating cells	[133]
human hepatocellular carcinoma cell line HepG2; datasets from the Oncomine platform	expressed, bradykinin increased TRPM7 expression	both channel and kinase functions were suggested	phosphorylation of non-muscle myosin heavy chain IIa and activation of calpains	TRPM7 increased migration and invasion abilities of tumor cells in response to bradykinin	[134]
human hepatoma cell lines, human HAP1 cells; mouse tumor xenografts	not investigated	channel-mediated Mg^2+^ influx and phosphorylation of RhoA by kinase	TRPM7/MRTF-A signaling pathway involving phosphorylation of RhoA by TRPM7 kinase, actin polymerization, and MTRF-A/Serum Response Factor target gene expression	blockade of TRMP7 resulted in oncogene-induced senescence and growth arrest of human hepatocellular carcinoma cells	[38]

**Table 6 ijms-25-00719-t006:** Role of the TRPM7 chanzyme in the pathophysiology of colorectal cancer.

Tumor Type/Experimental Sample	TRPM7 Expression	Channel/Kinase Function Involved	Observed/Implicated Signaling Mechanism	Major Finding	Ref.
human colon cancer LoVo cells	expressed in tumor cells; expression was downregulated in doxorubicin-resistant cells compared with that in doxorubicin-sensitive cells	not discussed	not investigated	Drug resistance was associated with alteration in magnesium homeostasis through modulation of TRPM7; its lower expression resulted in slower proliferation of resistant cells	[135,136]
human colorectal adenocarcinoma HT-29 and normal primary mouse colon epithelial cells; azoxymethane-induced colorectal cancer mouse model	expressed in tumor cells (was not compared with normal cells)	not discussed	not investigated	TRPM7 drove cancer cell proliferation; early-stage tumorigenesis was independent of systemic Mg^2+^ status and TRPM7	[137]
tissue samples from patients with colorectal cancer	overexpressed in human colon tumor samples compared with non-neoplastic tissues; adenocarcinomas showed a higher TRPM7 expression than adenomas	not discussed	not investigated	TRPM7 was overexpressed in colorectal cancer and expression positively correlated with tumor grade	[138]
Gene Expression Omnibus database, human colon cancer tissue and cancer cell lines	expression positively correlated with tumor infiltration, metastasis and clinical stage of colorectal cancer	not discussed	not elucidated	TRPM7 was involved in cell proliferation, migration and the invasion of colorectal cancer cells in vitro, and correlated with metastasis and clinical stage in vivo	[139]
The Cancer Genome Atlas	expression was higher in younger patients with rectal cancer	not discussed	not investigated	survival analysis showed no prognostic value for TRPM7 in colon and rectal cancers	[140]
human colonoscopy tissue samples	not investigated	not discussed	not investigated	TRPM7 gene polymorphisms, calcium/magnesium intake ratio and their balance influenced colorectal neoplasia risk	[141,142]
human colorectal carcinoma HCT116 cell line, mouse colon carcinoma CT26 cell line	Mg^2+^ deficiency with increased Ca^2+^/Mg^2+^ ratio resulted in increased TRPM7 expression	not discussed	calpain expression and activity were increased in response to Mg^2+^ deficiency with increased Ca^2+^/Mg^2+^ ratio	Mg^2+^ deficiency with increased Ca^2+^/Mg^2+^ ratio was associated with increased TRPM7 expression, increased cell migration and more aggressive, metastatic phenotype of colon cancer cells	[143]

**Table 8 ijms-25-00719-t008:** Role of the TRPM7 chanzyme in the pathophysiology of breast cancer.

Tumor Type/Experimental Sample	TRPM7 Expression	Channel/Kinase Function Involved	Observed/Implicated Signaling Mechanism	Major Finding	Ref.
human breast cancer cell line and breast cancer tissue samples	overexpression in cancerous tissues compared with normal; expression positively correlated with the mitosis marker Ki67	Ca^2+^ influx was suggested	not investigated	essential importance of TRPM7 channels for the proliferative potential of breast cancer cells	[155]
human breast cancer cell lines and breast cancer tissue samples	overexpressed in the poorly differentiated and highly proliferative tumor samples	not discussed	not investigated	TRPM7 was overexpressed in tumors with a high proliferative index	[67]
human breast cancer cell lines and breast cancer tissue samples; mouse xenograft model	expression was highest in high-grade primary tumors; expression associated with cancer progression and metastasis formation	phosphorylation by kinase was implicated	TRPM7 regulated myosin II–based cellular tension	TRPM7 promoted cell migration and metastasis formation; TRPM7 expression predicted poor outcomes	[156]
human breast tumor and lymph node tissue samples; human breast cancer cell lines	highly expressed in the invasive breast cancer epithelial cells in ER- breast tissues and metastatic lymph nodes	kinase-dependent mechanism was suggested	phosphorylation of myosin IIA	TRPM7 contributed to cell migration and was implicated in invasiveness as well as dissemination of tumor cells	[157]
human breast cancer cell line MDA-MB-435; datasets from the Oncomine database	expression was higher in metastatic cancers compared with that at the primary tumor site	increased intracellular Ca^2+^	Src-MAPK signaling pathway	TRPM7 enhanced migration and invasion	[158]
blood samples from women	not investigated	not investigated	not investigated	polymorphisms of TRPM7 gene were associated with breast cancer risk	[159]
human breast cancer tissues	highly expressed in the luminal A subtype of breast cancer	not investigated	not investigated	promoter methylation of TRMP7 was associated with better prognosis of disease	[160]
human breast cancer tissues	overexpression in breast carcinoma samples with microcalcifications compared with tumors without calcification	not discussed	not investigated	increase in expression of TRPM7 in infiltrating ductal carcinoma samples with microcalcifications	[161]
human breast cancer cell lines	time-dependent increase in TRPM7 expression following induction of mineralization	not discussed	not investigated	role for TRPM7 in promoting calcification of breast cancer cells	[162]
human breast cancer cell lines; HEK293 cells	expression was confirmed in breast cancer cells	not elucidated	not investigated	TRPM7 regulated cell cycle of breast cancer	[164]
human breast cancer cell lines	expression was confirmed in breast cancer cells	not elucidated	regulation of cyclin proteins	TRPM7 antagonism suppressed cell growth by arresting cell proliferation	[166]
breast cancer cell lines, HEK293 cells	expression was confirmed in breast cancer cells	not discussed	not investigated	TRPM7 was a target of lidocaine, for both viability and migration regulation	[165]
breast cancer cell lines, T-REx-293 cells	not tested	kinase activity was involved	phosphorylation of the myosin IIA heavy chain and of focal adhesion kinase	blockade of TRPM7 kinase suppressed breast cancer cell migration	[167]
human breast cancer cell line MDA-MB-468	expression was manipulated	effect was highly calcium-dependent and partially regulated by channel function	regulation of EGF-induced STAT3 phosphorylation and expression of the EMT marker vimentin	TRPM7 was a partial regulator of EMT in breast cancer cells	[168,169]
human breast cancer cell lines MCF-7, various other human cell lines, mouse cell line MEF	expression was manipulated	zinc-permeable channel function but not kinase domain was involved	increased cytosolic Zn^2+^ concentration modulated cellular levels of MDMX	TRPM7 regulated MDMX by modulating intracellular Zn^2+^ concentration to promote tumorigenesis (cell migration)	[170]
human breast cancer cell lines	expression was manipulated	TRPM7 channel function was suggested (Ca^2+^ influx)	increased Ca^2+^ and cellular FLICE-inhibitory protein	suppression of TRPM7 synergistically increased TRAIL-induced anti-proliferative effects and apoptosis	[171]
human breast cancer cell lines	expression was manipulated	phosphorylation by kinase was implicated	inhibition of myosin II function and cellular tension (part of a mechanical signaling hub that controls cell plasticity)	TRPM7 was required for maintenance of a mesenchymal phenotype of breast cancer cells	[172]

## Data Availability

Not applicable (new data were not created).

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
