# Peer review of "The Role of TRPM7 in Oncogenesis"

_ijms, 2024, doi:10.3390/ijms25020719_

Round 1
Reviewer 1 Report
Comments and Suggestions for Authors
The role of TRPM7 in various tumors in multiple organs and tissues has been described. In this review, TRPM7 is inspected in tumors of the nervous system, head and neck cancers, malignant neoplasms of the upper gastrointestinal tract, colorectal carcinoma, lung cancer, neoplasms of the urinary system, breast cancer, malignant tumors of the female reproductive organs, prostate cancer and other neoplastic pathologies.
The study has briefed on expression profiles, tumor hallmarks, and intracellular signaling pathways.
More than 9 different types of cancers have been considered. Appropriate references are provided. However, the roles of TRPM7 chanzyme in all these types of cancers could have been analyzed using appropriate diagrams representing various types of effects at the molecular levels besides presenting them using texts.
Let me copy from abstract 'Experimental and clinical data show that increased expression and/or function of TRPM7 are observed in most malignant tumor types.' Where are the clinical data? As it is a review article we need to see some briefs on them presented here taking permission from the original article that published them.
In the abstract, 3rd line: 'TRP' needs to replace 'TPR' superfamily
The conclusion contains no brief messages that might help readers to get some understanding. The conclusion needs to be expanded with more in-depth concluding remarks.
I hope these hints will help you develop your manuscript.
Author Response
Reply to Reviewer 1.
Thank you very much for your thorough and supportive review.
In response to your questions, advices/hints, and concerns about the manuscript, we have made the following corrections to the MS. (Changes are indicated with blue typing in our answer and in the MS).
„More than 9 different types of cancers have been considered. Appropriate references are provided. However, the roles of TRPM7 chanzyme in all these types of cancers could have been analyzed using appropriate diagrams representing various types of effects at the molecular levels besides presenting them using texts.”
In order to provide a more appropiate tumor-specific overview as you proposed, we replaced Table 1. with 10 more detailed and comprehensive Tables. In these Tables we systematically and comparably reviewed and indicated the relevant information about the TRPM7 expression pattern, implicated signaling pathway(s) and the major findings of the original publications.
„Let me copy from abstract 'Experimental and clinical data show that increased expression and/or function of TRPM7 are observed in most malignant tumor types.' Where are the clinical data? As it is a review article we need to see some briefs on them presented here taking permission from the original article that published them.”
The „clinical” word referred to examination of pathological tissue samples or public human databases in several original publications reviewed in this MS. In order to avoid misunderstanding, we deleted this word.
„In the abstract, 3rd line: 'TRP' needs to replace 'TPR' superfamily”
We corrected it.
„The conclusion contains no brief messages that might help readers to get some understanding. The conclusion needs to be expanded with more in-depth concluding remarks.”
We expanded the Conclusions.

Reviewer 2 Report
Comments and Suggestions for Authors
In the manuscript “The role of TRPM7 in Oncogenesis” by Köles and collaborators it is reviewed a collection of facts about the melastatin-transient receptor potential receptor 7 (TRPM7) expressed in a variety of cancers occurring in different human tissues, and the potential roles that this ion channel/kinase exerts in the oncogenic process. Despite the difficulty in integrating in a single picture the enormous collection of somewhat disconnected observations about the multiple roles that TRPM7 appears to exert in tumor cells, it seems that the presented information described in this manuscript could be of interest and useful to colleagues and students working in the field of channels, and also to oncologists, as it could stimulate further studies in this area.
There are some points that it could be of interest to clarify and others in need of correction, as it is detailed below.
1. Apparently, a major consensus in this field is that the overexpression of TRPM7 in tumor cells enhances cell proliferation. However, they refer to a case in esophageal carcinoma where TRPM7 induced a reduction of cell proliferation, in addition to inhibiting the migration and invasiveness of the tumor cells. Is TRPM7 overexpressed in these tumors or it just present normal expression patterns? It could be of interest to describe, if known, the implicated signaling pathway(s) and the mechanism(s) of such effects.
2. What is more relevant for the enhanced proliferative effect of tumor cells induced by TRPM7 overexpression, the expected increment in cytosolic Ca2+ concentration due to increased influx, or a higher phosphorylation level of the targeted proteins expected to occur via the Ser/Thr-kinase activity of the channel? Or are both activities important? If this is the case, why? It would of interest to discuss these points, if they are known.
3. It is not clear whether or not the expression and/or functions of TRPM7 in control neighboring normal cells in solid tumors was measured in all the reports mentioned in the review and whether or not differ from those in the cancerous cells under study.
4. When mentioning TRP-NOMPC (TRPN) it could be of interest to indicate that it refers to the “no mechanoreceptor potential C channel”.
5. In the phrase “…four subunits in homo- or tetrameric formation…” the word “OR” is confusing, as both concepts (homo and tetramer) are not necessarily incompatibles. Please clarify.
6. In the lower part of the drawing in Fig. 2, there is section referring to “Inducing or accessing vasculature” do you mean angiogenesis and extravasation of tumor cells across the endothelium? Please clarify.
7. In Table 1, “ER+ brest cancer” should be “ER+ breast cancer”. Please correct.
8. The phrase “Hyper-O-GlcNAcylation of caveolin-1 (Cav-1) and c-Myc was resulted in increased…” should be “Hyper-O-GlcNAcylation of caveolin-1 (Cav-1) and c-Myc resulted in increased…” Please correct.
9. the phrase At the end of the phrase “A Turkish retrospective study…” in parenthesis it is indicated “regardless of the atypical criteria”. Please clarify what those “atypical criteria” are.
10. In English “myeloma multiplex” should be “multiple myeloma”. Please correct.
11. The title of some articles in the Reference list appear in capital letters and in other articles no. Is this okay?
Comments on the Quality of English LanguageOnly minor corrections required.
Author Response
Reply to Reviewer 2
Thank you very much for your thorough and supportive review.
In response to your questions, advices/hints, and concerns about the manuscript, we have made the following corrections to the MS. (Changes are indicated with blue typing in our answer and in the MS).
- Apparently, a major consensus in this field is that the overexpression of TRPM7 in tumor cells enhances cell proliferation. However, they refer to a case in esophageal carcinoma where TRPM7 induced a reduction of cell proliferation, in addition to inhibiting the migration and invasiveness of the tumor cells. Is TRPM7 overexpressed in these tumors or it just present normal expression patterns? It could be of interest to describe, if known, the implicated signaling pathway(s) and the mechanism(s) of such effects.
In order to clarify this issue, we replaced Table 1. with 10 more detailed and comprehensive Tables. In these Tables we systematically and comparably reviewed and indicated the relevant information about the TRPM7 expression pattern, implicated signaling pathway(s) and the major findings of the original publications. The relevant aspects have been discussed more prominently in the conclusions as well.
- What is more relevant for the enhanced proliferative effect of tumor cells induced by TRPM7 overexpression, the expected increment in cytosolic Ca2+ concentration due to increased influx, or a higher phosphorylation level of the targeted proteins expected to occur via the Ser/Thr-kinase activity of the channel? Or are both activities important? If this is the case, why? It would of interest to discuss these points, if they are known.
This issue is also presented in the Tables and discussed in the Conclusions.
- It is not clear whether or not the expression and/or functions of TRPM7 in control neighboring normal cells in solid tumors was measured in all the reports mentioned in the review and whether or not differ from those in the cancerous cells under study.
This issue is also presented in the Tables and discussed in the Conclusions.
- When mentioning TRP-NOMPC (TRPN) it could be of interest to indicate that it refers to the “no mechanoreceptor potential C channel”.
We included it in the MS.
- In the phrase “…four subunits in homo- or tetrameric formation…” the word “OR” is confusing, as both concepts (homo and tetramer) are not necessarily incompatibles. Please clarify.
We corrected this mistake.
- In the lower part of the drawing in Fig. 2, there is section referring to “Inducing or accessing vasculature” do you mean angiogenesis and extravasation of tumor cells across the endothelium? Please clarify.
We revised Fig.2 and added a reference.
- In Table 1, “ER+ brest cancer” should be “ER+ breast cancer”. Please correct.
Table 1. was deleted and repleced with other Tables as indicated above.
- The phrase “Hyper-O-GlcNAcylation of caveolin-1 (Cav-1) and c-Myc was resulted in increased…” should be “Hyper-O-GlcNAcylation of caveolin-1 (Cav-1) and c-Myc resulted in increased…” Please correct.
We corrected it.
- the phrase At the end of the phrase “A Turkish retrospective study…” in parenthesis it is indicated “regardless of the atypical criteria”. Please clarify what those “atypical criteria” are.
We clarified the meaning of it.
- In English “myeloma multiplex” should be “multiple myeloma”. Please correct.
We corrected it.
- The title of some articles in the Reference list appear in capital letters and in other articles no. Is this okay?
It was generated by the EndNote program, now it seems to be fixed.

Round 2
Reviewer 1 Report
Comments and Suggestions for Authors
Well done. My previous concerns have been addressed. This article will serve as a quality reference review.
Reviewer 2 Report
Comments and Suggestions for Authors The authors have responded adequately and amply to all the points mentioned in my review by adding very detailed Tables and text in the Conclusions section.